# Targeting Breast Cancer-Derived Stem Cells by Dietary Phytochemicals: A Strategy for Cancer Prevention and Treatment

**DOI:** 10.3390/cancers14122864

**Published:** 2022-06-10

**Authors:** Kumari Sunita Prajapati, Sanjay Gupta, Shashank Kumar

**Affiliations:** 1Molecular Signaling & Drug Discovery Laboratory, Department of Biochemistry, Central University of Punjab, Guddha, Bathinda 151401, India; ksunita.prajapati@gmail.com; 2Department of Urology, Nutrition, Pharmacology and Pathology, Case Western Reserve University, Cleveland, OH 44106, USA

**Keywords:** breast cancer stem cells, dietary phytochemical, luminal A subtype, signaling pathway, cancer prevention, therapeutic resistance

## Abstract

**Simple Summary:**

Luminal A subtype breast cancer is the most prevalent form of breast malignancy with frequent diagnosis in women. Breast cancer stem cells (BCSCs) are a rare population of cells present therein that cause cancer aggressiveness, relapse, drug-resistance, poor therapeutic outcome and a decrease in overall survival of these patients. The published literature indicates that dietary phytochemicals have the potential to target stemness and self-renewal properties in luminal A-derived BCSCs. The aim of this review is to highlight the anticancer potential of dietary phytochemicals against luminal A-derived BCSCs and their underlying mechanism(s). These findings necessitate in-depth preclinical and clinical studies on phytochemicals to explore their role in breast cancer prevention and treatment.

**Abstract:**

Breast cancer is heterogeneous disease with variable prognosis and therapeutic response. Approximately, 70% of diagnosed breast cancer represents the luminal A subtype. This subpopulation has a fair prognosis with a lower rate of relapse than the other clinical subtypes. Acquisition of stemness in luminal A subtype modifies the phenotype plasticity to accomplish increased aggressiveness and therapeutic resistance. Therefore, targeting luminal A-derived breast cancer stem cells (BCSCs) could be a promising strategy for its prevention and treatment. Extensive studies reveal that dietary phytochemicals have the potential to target BCSCs by modulating the molecular and signal transduction pathways. Dietary phytochemicals alone or in combination with standard therapeutic modalities exert higher efficacy in targeting BCSCs through changes in stemness, self-renewal properties and hypoxia-related factors. These combinations offer achieving higher radio- and chemo- sensitization through alteration in the key signaling pathways such as AMPK, STAT3, NF-ĸB, Hedgehog, PI3K/Akt/mTOR, Notch, GSK3β, and Wnt related to cancer stemness and drug resistance. In this review, we highlight the concept of targeting luminal A-derived BCSCs with dietary phytochemicals by summarizing the pathways and underlying mechanism(s) involved during therapeutic resistance.

## 1. Introduction

Breast cancer is a heterogeneous disease that generally initiates in the milk glands or in ducts, which is referred to as lobular/ductal carcinoma in situ. These neoplastic breast cancer cells migrate through the gland/duct walls and proliferate in the surrounding tissue, which ultimately transform into an invasive phenotype. Breast cancer is frequently characterized by four major molecular subtypes that includes the luminal A (HR+/HER2−), luminal B (HR+/HER2+), basal-like (HR−/HER2−), and HER2-enriched (HR−/HER2+) subpopulations. The molecular subtypes (luminal A, luminal B, basal-like, and HER2 enriched) constitute approximately 73%, 11%, 12%, and 4% of breast cancer [1]. The other less common molecular subtypes include claudin-low and molecular apocrine forms. These molecular subtypes assist patient categorization, allowing them for better management of the disease as well as therapy-type decisions. Of the four major subtypes, luminal A tumors grow at a slower rate and are less aggressive with fairly high survival and low recurrence [1]. Conventional therapies target rapidly growing malignant cells that result in extensive elimination of tumors. Nonetheless, the surviving fraction constituting the minimal residual disease expands and undergoes multi-lineage differentiation to reconstitute the tumor. The reemerged tumor is highly aggressive, drug resistant, and comprises a new phenotype exhibiting increased aggressiveness and stemness. These cells are often referred as “cancer stem-like cells”. Targeting breast cancer stem cells (BCSCs) is the key to improving the efficacy of breast cancer. These cells have self-renewal properties and express stemness markers. BCSCs play an important role in the causation of drug resistance and results in a poor clinical outcome. Researchers are finding ways to target and remove the bulk of the tumor mass along with BCSCs for effective prevention and/or treatment of breast cancers. BCSCs play an important role in cancer metastasis due to the aberrant expression of some stemness-related factors, such as CD44, SOX2, OCT4, c-MYC, KLF4, Nanog, and SALL4 [2,3]. Besides the high expression of stemness and self-renewal markers, the aberrant expression of molecular signaling pathways including Wnt/β-catenin, Notch, Hedgehog, JAK-STAT, and PI3K/Akt/mTOR in BCSCs are shown to be involved in the pathophysiology of the disease [4,5]. Overall, the present literature reveals that the occurrence of BCSCs in the tumor microenvironment positively correlates with disease recurrence, low survival rate, chemo/radio therapy resistance, and low therapeutic output [6].

## 2. Clinical Characteristics of Breast Cancer Subtypes

Advanced molecular techniques provide a clear-cut characterization of breast cancer subtypes with better prediction and prognosis. Gene-based molecular assays, such as 70-gene and 80-gene signatures, 21-gene Recurrence Score, PAM50-ROR (50-gene Prediction Analysis of Microarrays and Risk of Recurrence), Endo-Predict, and the Breast Cancer Index (BCI) provide a precise characterization of breast cancer subtypes with better prediction and prognosis. The 21-gene recurrence score quantifies the probability of distant recurrence in patients with node-negative, ER+ breast cancer treated with tamoxifen into low, intermediate, and high-risk categories. The PAM50 classifies the intrinsic subtypes (luminal A, luminal B, HER2 enriched, basal, and normal breast) with risk of recurrence (low, intermediate, and high). The PAM50-ROR determines the probability of distant recurrence over 10 years. The 70-gene signature has the ability to divide patients into low- or high-risk corresponding to 10-year distant-metastasis-free survival (DMFS). The 80-gene signature divide patients into luminal, basal, and HER2 intrinsic subtypes. The combined 70- and 80-gene signatures are also able to classify breast cancer patients into luminal A-like (luminal subtype and low-risk), luminal B-like (luminal and high-risk), HER2, and basal subtypes [7,8].

Clinical reports suggest that at least half of the newly diagnosed breast cancer belongs to luminal A subtype. Initially, in 2011–2013, the oncologist proposed the molecular basis for the treatment of early breast cancer, which defined luminal A breast cancer patients as having a estrogen receptor (ER) positive, progesterone receptor (PR) positive (20%), HER2 negative, and Ki67 positivity of <14% [9,10]. This definition was based upon the gene profile (gene-based assay) and immunohistochemistry (IHC-based markers) of the tumor. The patients with IHC-based luminal A tumors were demonstrated to have better disease-free survival if PR expression was >20%. Especially for the luminal A breast cancer subtype, the patients and clinicians prefer surrogate IHC-based markers over gene-based biomarkers to establish the subtype. Overall, both the gene- and IHC-based biomarkers have been utilized for subtyping and treatment preference for breast cancer patients [11].

## 3. Relevance of Targeting Luminal A-Derived Breast Cancer Stem Cells

It has been widely accepted that the luminal cells of the mammary gland are unipotent, i.e., produce one cell type and have a self-renewal potential after evolution. Song et al. (2019) reported that during pregnancy or hormonal stimulation, the luminal cells give rise to luminal-derived basal cells (LdBCs) expressing the basal markers and ERα receptors. These cells respond to hormones and possess stem-cell renewal capability in the mammary gland [12]. Previously, it was reported that the Wnt and Notch signaling are determinants of the basal fate and luminal lineage. Recent findings align with the previous report that LdBCs demonstrated increased Wnt signaling (Figure 1). The plasticity of mammary luminal cells are associated with tumor progression in breast cancer patients [12]. The molecular subtype of breast cancer shows differential therapeutic response. Mei et al. (2020) studied the cancer stemness relationship with the molecular subtype of breast cancer in a comparative manner. The group developed MCF-7 cells (referred to as OKMS), which concomitantly overexpressed stemness-related genes (OCT4, KLF4, MYC, and SOX2) at the mRNA and protein level. These OKMS cells exhibited relatively low ER and higher HER2 expression. The cell growth and migration potential was increased up to 1.5 fold and the cancer stem cell population was increased up to 16% in OKMS cells, compared to parental MCF-7 cells. The OKMS cells demonstrated a drug response similar to HER2 positive cells after treatment with tamoxifen and trastuzumab. These results suggest a shift towards more aggressive and malignant tumors. However, luminal A cells exhibit slow growth and are less malignant but increase in stemness, transforming them to a more aggressive phenotype [13]. In another study, Yousefnia et al. (2019) reported that after a few passages, luminal A cells demonstrate a greater number of mammosphere formation in comparison to triple negative breast cancer cells [14]. Kim et al. (2012) demonstrated that tumor-derived pure luminal-like cells were capable of initiating invasive tumors, and generate larger tumors in comparison to basal-like cells in the in vivo model [15]. In addition, the stemness of cancer cells has the ability to change the tumor microenvironment in favor of supporting their aggressive natures [16]. Overall, these studies suggest that the stemness in the luminal A subtype transforms these cells to a more aggressive breast cancer phenotype. Since the luminal subtype forms the majority of breast cancer cases, targeting luminal A-derived BCSCs could provide a better therapeutic efficacy.

Epidemiologic and preclinical studies suggest that dietary phytochemicals possess chemopreventive properties against various cancer types. These phytochemicals possess anticancer, antioxidant, antiviral, antibacterial, and various other pharmacological activities. Dietary phytochemicals, such as curcumin, quercetin, resveratrol, silibinin, lycopene, and emodin have been found to be useful in decreasing cancer incidence. A number of studies summarize the anticancer potential of phytochemicals and their importance in drug development. Dietary phytochemicals have the ability to reduce cancer promotion by inhibiting cancer cell proliferation, survival, invasion/metastasis, angiogenesis, and eliminating toxic carcinogens from the body through targeting different cancer-related signaling pathways. Dietary compounds also regulate cell cycle progression, inflammatory cytokines, and oxidative stress response in cancer cells, and have the potential to modulate small non-coding RNAs’ expression alone or in a synergistic manner. More importantly, dietary phytochemicals have a greater ability to prevent/inhibit the formation of a small population of cancer cells often referred to as cancer stem-like cells by targeting various signaling pathways [17,18,19,20,21,22,23,24,25,26,27]. Therefore, targeting the luminal A-derived BCSCs by dietary phytochemicals could be a better strategy in the prevention and clinical management of breast cancer. The involvement of dietary phytochemicals in the management of BCSCs may provide significant contribution in this direction due to their minimal toxicity, low-cost, and bioavailability. In this article, we have reviewed the putative role of dietary phytochemicals in targeting luminal A-derived BCSCs in order to provide a strategy to increase the therapeutic efficacy in breast cancer patients. As most studies have focused on the role of dietary and non-dietary phytochemicals in cancer stem cell pathophysiology, to date, there is no comprehensive analysis to explore the role of dietary phytochemicals related to luminal A-cell-derived BCSCs. Here, we discuss the molecular mechanism(s) and pathways through which these phytochemicals target luminal A-cell-derived BCSCs. In addition, we also provide information related to pharmacokinetics, metabolism, bioavailability, dosage, and changes in the microbiome.

## 4. Source of Data

The literature published (in the English language) and indexed in the PubMed database was utilized for the present review. The relevant studies were retrieved through the use of “breast cancer stem cell, luminal A, mammosphere” as keywords in searches of the database. The literature that contained the luminal A breast cell-derived cancer stem cells and phytochemicals was filtered. Various in vitro luminal A breast cancer cells (BT483, CAMA1, EFM19, HCC1428, HCC712, IBEP2, KPL1, LY2, MCF-7, MDAMB134, MDAMB134VI, MDAMB175, MDAMB175VII, MDAMB415, T47D, ZR751, and ZR75B) were taken into consideration; however, most of the studies were focused on the MCF-7 cell-derived mammosphere, which demonstrated increased stemness/self-renewal markers. Further, the literature was filtered on the basis of dietary and non-dietary phytochemicals by finding the associated published information (Table 1). Data published on the role of dietary phytochemicals on luminal A-derived breast cancer stem cells were only considered in the study. Dietary phytochemicals viz. curcumin, naringenin, resveratrol, genistein, quercetin, silibinin, and thymoquinone, comparatively cited in the literature, were used in the review. Additional phytochemicals such as ginsenoside Rg3, hisperidin, pristimerin, pterostilbene, 3-O-(E)-p-Coumaroyl betulinic acid and withaferin A, which are less studied, are discussed under a common heading, namely “other dietary phytochemicals”.

## 5. Curcumin and Stemness in BCSCs

Curcumin belongs to the polyphenolic group of phytochemicals and is an important component of the spice turmeric all around the globe, especially in India. It is used for its coloring ability and taste. Moreover, curcumin has been used in various medicinal preparations of Ayurveda and Chinese Medicinal systems. Numerous studies report potent biological activity, and applications in the food, biotechnology, and cosmetics industry. Curcumin is well-known for its health promoting and disease preventive properties [41]. Yang et al. (2020) studied the effect of curcumin (alone and in combination with nanoparticles) in the radiation-treated (4Gy) MCF-7 cell-derived mammosphere. The group found that curcumin significantly increased the radio-sensitivity in the breast sphere cells alone and in combination with the gold nanoparticles. More than 60% of the sphere population was significantly damaged by curcumin treatment. Curcumin pretreatment in the radiation-exposed breast cancer MCF-7 and MDA-MB-231 cell-derived mammosphere demonstrated increased apoptosis and reactive oxygen species formation, G0/G1 phase cell cycle arrest, and decreased HIF-1α and HSP90 protein expression [42]. Sarighieh et al. (2020) tested the efficacy of curcumin in cancer stem cells isolated from MCF-7 cells. The cells were sorted in the presence of CD44+/CD24− surface markers, and the curcumin treatment was rendered both in hypoxic and normoxic conditions. The curcumin-treated MCF-7-derived cancer stem-like cells demonstrated early apoptosis and G2/M phase arrest under hypoxic conditions. Under normoxic conditions, the curcumin induced S and G2/M phase arrest in the cells [43]. The curcumin inhibits HIF1 nuclear translocation by degrading ARNT (aryl hydrocarbon receptor nuclear translocator), which is required for the transcription of hypoxia-related genes in breast cancer stem cells. The treatment also decreases HIF1 and HIF-2α expression in MCF-7 cell-derived mammospheres. In another study, Borah et al. (2020) demonstrated that curcumin in combination with GANT61 (Gli1 and 2 inhibitor of hedgehog signaling pathway) loaded PLGA (poly(lactic-co-glycolic acid) nanoparticles demonstrated self-renewal inhibitory potential in MCF-7 cell-derived mammospheres [44]. Attia et al. (2020) reported the effect of curcumin in combination with paclitaxel and vitamin D on MCF-7 cells. The study demonstrated that it improved anticancer activity and anti-drug resistance efficacy in MCF-7 cells by a decrease in proliferation and increased apoptosis. Further treatment significantly decreased cancer stemness markers (multidrug resistance complex and aldehyde dehrogenase-1) at the protein level [45]. Liu et al. (2019) studied the effect of a curcumin coating on a synthetic polymer (oligomeric hyaluronic acid-hydrazone bond-folic acid-biotin), which forms curcumin nano-actiniaes (Cu-NA). The Cu-NA demonstrated increased toxicity in MCF-7 cells, breast cancer stem cells, and in the in vivo anticancer experiment in comparison to free curcumin and/or other control groups [46]. Hu et al. (2019) demonstrated that curcumin has the potential to inhibit the cancer stemness property (Oct4, Nanog, and Sox2) and the epithelial to mesenchymal (EMT) transition in luminal A cells. The group isolated the CD44+CD24−/^low^ subpopulation of MCF-7 cells, which shows breast cancer stem-like properties. The study reported that curcumin treatment resulted in a decreased cell proliferation and colony formation potential in these cells [47]. Hashemzehi et al. (2018) demonstrated the efficacy of a novel curcumin formulation (phytosomal encapsulated curcumin) on thrombin-induced cellular proliferation and metastasis in MCF-7 cells. The study demonstrated that curcumin formulation exerts its anticancer, anti-metastatic, and anti-stemness potential by activating AMPK signaling [48]. Li et al. (2018) demonstrated that curcumin treatment decreases the expression of stemness markers (ALDH1A1, CD44, Nanog, and Oct4) and inhibit Sonic hedgehog and Wnt signaling pathways in MCF-7 cell-derived mammospheres [49]. In another study, the folate decorated curcumin-loaded nanostructured lipid carriers (FA-CUR-NLCs) formulation demonstrated enhanced antitumor activity (in comparison to standard curcumin) in MCF-7 cell-inoculated animal models [50]. Yuan et al. (2018) studied the effect of the curcumin and doxorubicin combined nano-formulation (CURDOX-NPs) against drug resistance MCF-7 cell (MCF-7/ADR)-derived mammospheres. The study reported that the CURDOX-NPs significantly reduced the mammosphere formation potential in vitro and demonstrated tumor growth regression (~33%) in a mouse xenograft model [51]. Zhou et al. (2017) showed that curcumin has the potential to inhibit the drug resistance potential in breast cancer stem-like cells by improving Bcl-2-mediated apoptosis. The apoptosis inducing efficacy of curcumin was more pronounced during the combined treatment with Wnt and PI3K inhibitors. The treatment significantly decreased the anti-apoptotic proteins and increased the expression of pro-apoptotic proteins in targeted cells [52]. In a different study, Zhou et al. (2015) reported that curcumin has the potential to inhibit the cancer stem cell self-renewal potential in MCF-7 cell-derived mammospheres by lowering the expression of drug efflux transporters (ABCG2). Curcumin treatment also improved the anti-cancer efficacy of mitomycin C in the in vitro breast cancer stem cell model [53]. Further, Zhou et al. (2011) d that the co-treatment of curcumin and mitomycin C significantly reduced the mitomycin C-associated side-effects and improved its efficacy in a breast cancer xenograft model. Curcumin co-treatment altered the creatinine/blood urea nitrogen level and glutamic oxalacetic transaminase/glutamic pyruvic transaminase activity towards normal levels. The study indicated that curcumin mitigates the kidney reacted toxicity due to mitomycin C treatment in MCF-7 xenograft models. Moreover, curcumin and mitomycin C synergistically induced cell cycle arrest via the p38 MAPK pathway in the in vivo model compared to respective alone treatment(s) [54].

Other studies indicate that STAT3 (Signal Transducer and Activator of Transcription 3) and NF-κB (Nuclear factor-κB) signaling pathways play an important role in the maintenance of cancer stem-like properties in cancer cells. Chung and Vadgama, (2015) studied the effect of curcumin and EGCG co-treatment in CD44+ MCF-7-derived BCSCs. The study demonstrated a significant decrease in stem cell population, and decreased STAT3 phosphorylation and STAT3-NFĸB interaction in the curcumin-treated cells [55]. Furthermore, curcumin co-treatment with interferon-β/retinoic acid (IFN-β/RA) in the MCF-7 athymic nude mouse model demonstrated synergistic anticancer potential. Curcumin increases the IFN-β/RA level with a simultaneous decrease in cyclooxygenase-2 (COX-2) activity and increases the DNA damage-inducible gene 153 (GADD153) expression, resulting in reduced tumor growth [56]. A summarization about the curcumin potential against luminal A-derived breast cancer stem cells includes (i) the decrease in self-renewal/stemness marker(s) expression, (ii) the possessed radio-sensitization and chemo-sensitization potential, (iii) targets AMPK, STAT3, and NF-ĸB signaling pathways and hypoxia-related markers, (iv) decreases the expression of multi-drug resistance transporters (ABCG2), and (v) the possessed synergetic anti-breast cancer stem cell potential. The overall mechanism(s) of the curcumin regulation of breast cancer stemness and associated signaling pathways is shown in Figure 2.

## 6. Phytoestrogen and BCSCs

Phytoestrogens are polyphenolic naturally occurring secondary metabolites of plants. These molecules are structurally and functionally similar to the mammalian major sex hormone, 17-β-oestradiol (E2). Coumestans and isoflavones are widely researched phytoestrogens. These compounds are found in a variety of foods and possess a protective effect against hormone-related cancers and other diseases. The literature search related to the present review demonstrated that naringenin, resveratrol, genistein, apigenin, and quercetin possess the potential to target luminal A-derived BCSCs.

### 6.1. Genistein

Genistein is a naturally occurring isoflavone present in various dietary sources, including legumes and soybean products [57]. A hydroxyl group at carbon 7 forms a glycosidic linkage with sugar molecules and generates its dietary carbohydrate conjugates [58]. Genistein elicits several pharmacological activities, such as protein tyrosine kinase and DNA topoisomerase II inhibition, and the induction of antioxidant enzymes. The report demonstrates the role of genistein in the induction of apoptosis, cell cycle arrest, cell proliferation and metastasis inhibition in breast cancer experimental models. Genistein alters the expression of cancer-associated signaling pathways alone or in combination with standard chemotherapeutic drugs [43]. The Krüppel-like factor 4 (KLF4), an evolutionary conserved zinc finger-containing transcription factor, regulates cellular proliferation. In cancer, it functions in an organ-specific manner and displays both an oncogenic and tumor suppressive role. KLF4 is highly expressed in >70% breast cancer patients, human breast cancer cell lines, and mouse mammary cell-derived stem cells. A high KLF4 expression was positively associated with the increased self-renewal/stemness markers and side-population in experimental settings [59]. A later report demonstrates that genistein in combination with sulforaphane significantly reduces KLF4 expression at mRNA and protein levels in MCF-7 cells [60]. BCSCs undergo increased endocytosis (clathrin and caveolin independent) than analogous non-stem cancer cells. Genistein was unable to inhibit the endocytosis in the MCF-7-derived mammosphere but demonstrated a mammosphere reduction potential [61]. Genistein inhibit mammosphere formation in MCF-7 cells by downregulating the Hedgehog and PI3K/Akt pathway and inhibiting abiogenesis in mammary glands [62,63,64]. Further, it has been reported that genistein target adipogenesis and stem cell formation in an interleukin-6 (IL-6) independent manner [64]. In a similar study, genistein at a 25 µM dose increases the sphere formation potential in MCF-7 cell-derived tertiary spheroids. The genistein increased protease inhibitor 9 (PI-9, granzyme b inhibitor) expression and decreased estrogen receptor isoform (ERα66) [65]. It also reduced the mammosphere formation potential in MDA-MB-231 cells both in co-cultured with MCF-7 cells and in solo culture at micro and nanomolar concentrations. Although the size of the spheroids was reduced in both cultures, the morphological changes were observed only in the co-cultured experimental setup. Another study revealed that genistein inhibits the stem cell formation in MDA-MB-231 cells via induction PI3K/Akt and MEK/ERK signaling pathways in a paracrine manner through the increased expression of amphiregulin in MCF-7 cells [66]. Genistein alone or in combination with other phytoestrogens has been reported to inhibit/reduce tumor growth both in in vitro and in vivo systems. The combination of genistein (GEN) and lignan enterolactone (ENL) (100 mg/kg ENL + 100 mg/kg GEN) demonstrated that the inhibition of estradiol (E2) induced in MCF-7 cells established tumor growth and angiogenesis in mice [67]. The published literature thus far suggests the effect of genistein against luminal A-derived BCSCs through the (i) downregulation of the Hedgehog and PI3K/Akt pathway, (ii) suppression of the oncogenic transcription factors and (iii) inhibition of the mammosphere formation in ER^−^ breast cancer cells in a paracrine manner.

### 6.2. Naringenin and Resveratrol

Naringenin belongs to flavanones, a subclass of flavonoid. The compound is widely distributed in tomatoes, citrus, and other fruits. It is also found in glycoside form, known as naringin in various dietary sources. Naringenin is insoluble in water and soluble in organic solvents. The compound possesses antioxidant, anti-inflammatory, anti-aging, anticancer, anti-asthma, and anti-viral properties. Moreover, it has been utilized in infertility, immuno-depression, constipation, hepatic damage, pregnancy, and obesity as a therapeutic molecule [68]. Naringenin has been reported to possess anticancer activity, induce apoptosis, and cell cycle arrest at different concentrations in various human breast cancer cell lines. Naringenin has been well documented in managing the cancer cell growth and proliferation, migration, and multi-drug resistance both in in vitro and in vivo by targeting signaling pathways such as Jak/Stat3, Notch1, p38/MAPK, NF-ҡB, PI3K/Akt, and COX2. Nanoparticle formulations of naringenin improves chemosensitization and anticancer potential [68]. Curcumin-naringenin loaded dextran-coated magnetic nanoparticles (CUR-NAR-D-MNPs) in combination with radiotherapy has been studied in the MCF-7 cell-inoculated mouse model. CUR-NAR-D-MNPs, in combination with radiotherapy, reduced tumor volume and induced cell cycle arrest and apoptosis by modulating the expression of p21, p53, TNF-α, CD44, and ROS signaling in experimental animals [69]. The group also studied the effect of phytoestrogens (naringenin, resveratrol, and quercetin) in the MCF-7 cell-derived xenograft model. The findings revealed that the phytoestrogen treatment inhibit the survival of breast tumor initiating cells, and restrict tumor growth rates and tumor initiation by increasing DAXX protein levels. Phytoestrogens demonstrated DAXX-mediated anti-breast cancer activity in the order of naringenin > resveratrol > quercetin at 20 mg/kg dose [70].

Resveratrol is a lipid soluble polyphenolic compound. The phytochemical is found in cis and trans conformations in dietary sources. The dietary source of resveratrol includes cranberry, red/white grapes, strawberry, peanuts, etc. Resveratrol has been well documented for its cardio-protective and cancer preventive properties [71]. After hormonal therapy, the tumor initiating cells in ER+ breast cancer typically demonstrate increased Notch signaling activity and resistance to therapy. These cells do not express the death domain-associated protein 6 (DAXX), resulting in the activation of notch signaling and formation of therapy-resistant tumor initiating cells (TICs) [70]. It is quite interesting that the treatment of MCF-7 cell-derived TICs with phytoestrogens (such as naringenin and resveratrol) demonstrated increased DAXX expression, which results in the inhibition of the Notch signaling-mediated TIC-cell enrichment. It should be noted that naringenin and resveratrol demonstrate this effect by selectively targeting both ERα and ERβ forms of the estrogen receptor. The findings of Peiffer et al. (2020) indicate that the selective inhibition of ER receptor isoforms by phytochemicals could be a novel therapeutic approach to target cancer stemness and tumor initiation in therapy resistant ER+ positive cells [70].

### 6.3. Quercetin

Quercetin (3,5,7,3′,4′-pentahydroxylflavone) belongs to a flavonol subclass of flavonoid. It is found in various fruits and vegetables, but at a larger quantity in apple and onion. Quercetin protects body tissues alone or in combination with other dietary antioxidants, such as vitamin C/E and carotenoids. The phytochemical is well known for its beneficial role in human health, as it is a powerful antioxidant, lowers cholesterol and erectile dysfunction, improves blood circulation, reduces inflammation, lowers the risk of neurodegenerative disease, and possesses anticancer properties [72]. Quercetin-3-methyl ether (Q3ME) is a natural analog of quercetin found in various plants. 7-O-geranylquercetin is an alkylated derivative of quercetin synthesized to overcome the poor solubility of quercetin. The compound possesses potent anti-tumor activity. It has been demonstrated that the derivative treatment in MCF-7/ADR cells inoculated in BALB/c nude mice significantly reversed drug resistance by down-regulating the expression of P-gp protein and its encoding gene MDR1 [73]. Cao et al. (2018) reported that quercetin has the potential to target MCF-7 cell-derived BCSCs alone or in combination with DAPT, a γ-secretase inhibitor. Q3ME inhibits the mammosphere formation by regulating the expression of genes involved in cancer stemness and inhibited the Notch and PI3-AKT signaling pathways [74]. The increased expression of P-glycoprotein (membrane transporter) mediated multidrug resistance (MDR) in cancer cells play a major role in decreasing the therapeutic outcome in breast cancer patients [75,76,77]. In this context, nuclear translocation of YB-1, a oncogenic transcription factor, is in association with the P-gp overexpressed in breast cancer cells and is associated with the stemness property [78]. Li et al. (2018) demonstrated that quercetin decreased YB-1 translocation, P-gp expression in luminal A breast cancer cells, and in the drug-resistant MCF-7 cell-derived stem-like cell population [79]. Similarly, Li et al. (2018) reported that quercetin, in combination with doxorubicin, significantly decreased the MCF-7-derived breast cancer stem-like cells [80]. Further, Li et al. (2018) demonstrated that quercetin significantly decreased the clone and sphere formation potential in MCF-7 cell-derived BCSCs [81]. Quercetin also modulated the PI3K/AkT/mTOR (phosphatidylinositol-3-kinase/Akt/mammalian target of rapamycin) signaling pathway in these cells [82]. Earlier, Imai et al. (2012) reported that the quercetin derivative (LY294002) has the capability to reduce the drug efflux in BCSCs by inhibiting the PI3K/Akt signaling pathway [83]. A summarization of the literature on the anticancer effects of quercetin against luminal A-derived breast cancer stem cells includes (i) targeting the ERα receptor, Notch and PI3/AKT/mTOR signaling pathways, (ii) lowering the expression of the multidrug resistance transporter, and (iii) inhibits the nuclear translocation of proteins involved in the breast cancer stemness and self-renewal process.

### 6.4. Silibinin

Silibinin is a natural flavonolignan compound. It is a major constituent of Silymarin, extracted from *Silybum marianum* (milk thistle), which is used for the treatment of liver diseases. Recently, silibinin has been reported to exert significant anti-neoplastic effects against breast, prostate, lung, colon, and skin cancer in the in vitro and in vivo model. It has been reported that metabolic heterogeneity plays an important role in cancer stem cell maintenance and self-renewal. Bonuccelli et al. (2017) found that MCF-7 cells possess high PGC1α activity and reactive oxygen species production, and NADH levels in the mitochondria possess increased stem cell formation potential. Silibinin significantly reduced the mammosphere formation in this metabolic form of MCF-7 cells [84]. Studies have indicated that silibinin is a glycolytic inhibitor and also inhibits glucose uptake. Dadras et al. (2016) studied the MCF-7 cells stemness inhibition potential of silibinin-entrapped nanoparticles. The study demonstrated a significant reduction in the potential of free silibinin in MCF-7 cancer stem cell viability in the in vitro assay [85]. Published reports on the efficacy of silibinin in luminal A-derived breast cancer cells are at the preliminary levels. There is a need to explore the underlying mechanism of stemness and the self-renewal property inhibition potential of silibinin in both the in vitro and in vivo models. The literature so far has demonstrated that silibinin exerts its pharmacological effect on luminal A-derived breast cancer stem cells by targeting metabolic pathways, and reducing the viability of MCF-7 cell-derived mammospheres.

### 6.5. Thymoquinone

*Nigella sativa* L. (Ranunculaceae), commonly known as black cumin, is an important spice in Europe, South West Asia, and North Africa. Thymoquinone (TQ) belongs to the quinone group of phytochemicals and is the most abundant constituent of *Nigella sativa* seeds in volatile oil. Several pharmacological activities of TQ have been reported, including anti-histaminic, anti-inflammatory, anti-microbial, antioxidant, immunomodulatory, and anti-tumor effects. Various studies demonstrate that TQ possesses less toxicity and has low adverse effects on normal cells [86]. TQ demonstrated reduced self-renewal properties in the MCF-derived mammosphere. The TQ treatment decreased the number and size of spheroids in comparison to the control group. TQ, alone or in combination with the emodin, a natural anthraquinone derivative, lowered the cancer stemness-related markers viz. OCT-4, SOX-2, NANOG, and ALDH1/2 in MCF-7-derived BCSCs [87]. In another study, the TQ decreased the stem cell population by 12%; in combination with paclitaxel, the efficacy was increased up to 32%, as compared with the non-treated cells. Whereas, the paclitaxel treatment alone demonstrated an 8% reduction in stem cell population. In another study, the gemcitabine was unable to decrease the stem cell population; however, in combination with TQ, a 12% reduction was observed [88]. These studies indicate that TQ may be used in combination with standard chemotherapeutic drugs to target BCSCs effectively. However, detailed mechanisms are warranted to establish the BCSCs’ reduction potential of TQ, in combination with the standard therapeutic drugs in pre-clinical models, including a reduction in mammosphere size and number alone or in combination with a phytochemical, as well as the chemosensitization potential of BCSCs. The luminal A-derived mammosphere reduction and breast cancer stemness and self-renewal regulating pathways’ inhibiting potential of phytoestrogens are summarized in Figure 3.

## 7. Other Dietary Phytochemicals

Hisperidin is a natural polyphenolic compound generally known as a citrus flavonoid. The compound is pharmacologically active and possesses significant anticancer activity. Hesperidin and its aglycone derivative (hesperitin) inhibit metastasis and tumor growth in MCF-7-inoculated experimental mice. Hesperitin (at a 1000 and 5000 ppm concentration) prevent tumor growth by reducing the plasma estrogen level and pS2 gene expression in the ovariectomized, and the aromatase overexpressing in the MCF-7 athymic xenograft mouse model. The hesperitin treatment inhibited aromatase activity and increased the cell cycle arrest and apoptosis in experimental animals [89]. Recently, Hermawan et al. (2021) reported the anticancer activity of hesperidin in luminal A-derived breast cancer spheroids using bioinformatics and an in vitro approach. A decreased sphere and colony formation potential was observed in hesperidin-treated MCF-7 cells [90].

Pristimerin is a natural occurring triterpenoid compound mainly isolated from the *Celastraceae* and *Hippocrateaceae* family. It exerts anticancer activity against different types of cancer both in in vitro and in vivo experimental models. Cevatemre et al. (2018) demonstrated that pristimerin inhibited breast tumor growth and induced apoptosis by cleaving PARP and activating caspase-3 in MCF-7-derived spheroids and in the mouse xenograft model [91]. Pristimerin demonstrated increased apoptosis, cytoplasmic vacuolation, endoplasmic reticulum stress, unfolded protein response, and autophagy flux blockage-mediated death in breast cancer spheroids. The phytochemical inhibited the Wnt signaling pathway in MCF-7-derived mammospheres by degrading the low-density lipoprotein receptor-related protein 6 (LRP6), a Wnt co-receptor [91].

Pterostilbene is a natural dimethylated analogue of resveratrol. The compound is present in blueberries in larger quantity. Pterostilbene inhibits cancer cell proliferation, invasion, and metastasis, and induced apoptosis in several experimental models. Mak et al. (2013) studied the effect of pterostilbene in M2 tumor-associated macrophages (M2TAMs) inducing stem cell generation potential in MCF-7 cells. The pterostilbene treatment significantly reduced the BCSC generation in MCF-7 cells co-cultured with the M2TAMs by decreasing the CD44+/CD24− cell population, and reduced the migratory and invasive capabilities of BCSCs. Moreover, the pterostilbene treatment reduced NF-κB, vimentin and Twist1, and elevated the E-cadherin expression in MCF-7-derived mammospheres [92]. Later, Wu et al. (2015) reported that pterostilbene produced more toxicity in MCF-7-derived cancer stem cells in comparison to MCF-7 cells. Pterostilbene significantly induced necrosis-mediated cellular membrane damage, reduced stemness markers CD44 and c-Myc, and inhibited hedgehog, Akt and GSK3β signaling pathways in MCF-7-derived spheroids [93].

Ginsenoside Rg3 (GRg3), which belongs to the *Araliaceae* family, is an important pharmacological active constituent of *Panax ginseng*. GRg3 has been known to possess potent anticancer activity by modulating several oncogenic pathways. Oh et al. (2015) comparatively studied the efficacy of low and high GRg3 containing red ginseng extract in the MCF-7 cell-derived mammosphere [94]. The high GRg3 content extract demonstrated more pronounced effect on the mammosphere by decreasing their self-renewal potential. In a different study, GRg3 decreased the stemness and self-renewal potential in MCF-7 cells’ spheroids by targeting the PI3K/Akt signaling pathway, modulating Sox-2 and Bmi-1 self-renewal markers, and inhibiting the nuclear translocation of the HIFα factor [95].

Withaferin A (WA) is a major pharmacological active ingredient of Indian Ginseng (*Withania somnifera*). It belongs to the steroidal lactone group of compounds. WA inhibits luminal A-derived BCSCs by reducing the urokinase type plasminogen activator receptor (uPAR) and polycomb group protein Bmi-1 expression, which are well known factors to drive stemness and self-renewal properties in BCSCs. Inhibition of Notch4 activity reduces stem cell activity and tumor formation both in in vitro and in vivo breast cancer models [96]. Kim and Singh (2014) reported that WA has the potential to inhibit Notch4 activation and reduce KLF4 expression and ALDH1 activity in MCF-7 and SUM159 cancer stem cells [97]. The modulation of miRNAs by phytochemicals is an important strategy to modulate the expression of target mRNAs at a transcriptional level in cancer cells [13,98]. Higher expression of miR-6844 has been demonstrated in clinical specimens of invasive breast cancer patients in comparison to normal subjects. Recently, our group reported that miR-6844 is highly expressed in the luminal A-cell-derived mammosphere, compared to normal non-sphere breast cancer cells. The WA treatment significantly reversed the expression of miR-6844 and inhibited the mammosphere formation potential [99]. The aberrant expression of Notch signaling target genes has been positively correlated with the stemness and self-renewal property of BCSCs. Recently, we demonstrated that natural phytochemical 3-O-(E)-p-Coumaroylbetulinic acid possesses stemness and self-renewal inhibition potential in the MCF-7 cell-derived mammosphere by downregulating the Notch signaling pathway. The phytochemical altered the Notch target genes such as E-cadherin, Hey1, and Hes1, and breast cancer stemness markers viz. c-Myc, COX2, CD44, OCT4, NANOG, CD44, and EpCAM in MCF-7 cell-derived mammospheres [100].

Overall, the phytochemicals discussed above possess the potential to target luminal A-derived BCSCs by the selective elimination of these cells through the inhibition of Akt, Wnt, hedgehog/GSK3β signaling pathway, and nuclear translocation of the HIFα factor. These underlying mechanisms of the luminal A-derived breast cancer stemness and self-renewal inhibition potential of dietary phytochemicals is summarized in Figure 4 and Table 2.

## 8. Pharmacokinetics and Bioavailability

Liver is the primary site of metabolism for phytochemicals, together with the intestine and gut microbiota. After several metabolic reactions, phytochemical structures are modified in the hepatocytes and enterocytes, and form subsequent inactive/bioactive metabolites. Following oral ingestion, phytochemicals undergo extensive metabolism that includes chemical reactions including reduction, sulfation, and glucuronidation in the liver, kidneys, and intestinal mucosa. These metabolized products generated in the liver and intestine, as a result of their biotransformation, exhibit enhancement in their biological activity. Studies have demonstrated that the human microbiota and brush border enzymes are involved in phytoestrogen metabolism and active metabolite synthesis. Gut microbiota composition influences phytochemical bioavailability and inter-individual effects. Gut bacteria, for example, convert genistein, soy’s most abundant isoflavone, into several metabolites that target and alter estrogen-dependent and non-estrogenic pathways with a variety of biological activities. As such, individual differences in the microbiome, therapeutic possibilities, and anticancer effects may be unique to each person [101,102,103]. Overall, these variables hinder the clinical development of dietary phytochemicals. Preclinical studies suggest that the intestinal P-glycoprotein efflux pump is responsible for the limited bioavailability of the phytochemicals. Blocking ABC transporters with quercetin increases the bioavailability and decreases their efflux. Other studies demonstrate that the combination of dietary phytochemicals such as curcumin and piperine exhibits greater bioavailability. Reports on humans suggest that the intake of nonsteroidal anti-inflammatory agents, reserpine, and blood thinners exhibit adverse effects taken together with phytochemicals [102]. Efforts are also directed to enhance the bioavailability of phytochemicals to exhibit better preventive and/or therapeutic responses. In this direction, the high-bioavailability formulation of phytochemicals is under development [104,105]. Food processing such as heating, drying, and grinding, and factors such as climate change and plant stress could impact the bioavailability of dietary phytochemicals, having an effect on their biological activity. Technological strategies such as nanoparticle synthesis, phytochemical(s) in lecithin, the phosphatidylcholine carrier, and solid lipid nanoparticles demonstrate an increase in the bioavailability of dietary phytochemicals [105,106,107].

In preclinical studies, phytochemicals exhibit an antitumor potential at the doses 1–200 µM and 2–100 mg/kg body weight without any apparent toxicity utilized in in vivo models [54,56,73,89]. In recent studies, it is noted that dietary phytochemicals/phytoestrogen exhibit a significant reduction in tumor volume in the MCF-7-inoculated mouse xenograft model at 2–100 mg/kg body weight [54,56,73,89]. These doses can be extrapolated in studies on humans as well. Moreover, curcumin (500 mg BID; NCT01740323), genistein (100 mg; NCT00244933), resveratrol (474 mg phenolics/day; NCT03482401), ginsenoside Rg3 (20 mg BID; NCT01717066), and thymoquinone (500 mg; NCT04852510) have been studied in various clinical trials in the range of previously reported in vivo dosages or at higher dosages. These clinical trials on dietary phytochemicals either alone or in combination with standard chemotherapy demonstrated encouraging results in breast cancer patients with less toxicity and lower side effects (NCT03072992).

## 9. Study Strengths and Limitations

Few studies have summarized the role of dietary phytochemicals in targeting BCSCs. Some other available reviews related to this subject are discussed under cancer stem cell physiology and related hallmarks, and their modulation by phytochemicals. The present review summarizes the role of dietary phytochemicals and their effect on luminal A-derived BCSCs’ pathophysiology at the molecular level. The review emphasizes that dietary phytochemicals possess increasing potential to target breast cancer subtype (luminal A) cell-derived stem cells. This review also highlights the efficacy of dietary phytochemical-derived nanoparticles, and co-treatment with other drugs against luminal A-derived BCSCs. However, there are some limitations in the study, such as the lack of the extensive information on the effect of dietary phytochemicals in luminal A cells derived in in vivo experimental models. Because the luminal A-derived stem cell formation is an important event in the breast tumor microenvironment, further detailed studies are required to assess the stemness initiation, self-renewal, and chemotherapy resistance potential of dietary phytochemicals.

## 10. Conclusion and Future Prospects

Breast cancer luminal A subtype possess estrogen receptor and thus are responsive to hormone therapy. The luminal A subtype constitutes approximately 80% of the total breast cancer cases and demonstrates better prognosis and therapeutic susceptibility. Luminal A-derived T47D and MCF-7 cell culture models form tight three-dimensional cell–cell adhesion structures in comparison to luminal B breast cancer cells. As the breast tumor microenvironment constitutes nearly all subtypes of cancer cells, out of these, luminal A cells are slow growing and have the capability to transform into other breast cancer subtypes in response to changes in hormone levels and physiological states, including pregnancy status and various therapeutic modalities. Targeting luminal A cells at the initiation of breast cancer with phytochemicals could inhibit cancer progression and minimizes the casual switching to various subtypes. The dietary intake of these phytochemicals could offer prevention towards initiation and development of breast cancer, whereas cotreatment with standard chemotherapeutic drugs has the potential to increase the efficacy and therapeutic response. The present review highlights that dietary phytochemicals have the potential to target luminal A-derived BCSCs by lowering their stemness and self-renewal properties. These dietary phytochemicals demonstrate their anticancer effects by the modulation of signaling pathways, including AMPK, STAT3, NF-ĸB, Hedgehog, PI3K/Akt/mTOR, Notch, GSK3β, and Wnt, and other, as well as via the regulation of the mechanism(s) involved in the process of proliferation or drug resistance. These phytochemicals have the ability to target puttive molecular and biochemical events in luminal cell-derived mammospheres. The present review necessitates in-depth preclinical and clinical studies on dietary phytochemicals alone or in combination with the standard treatment to explore their cancer prevention and treatment potential.

## Figures and Tables

**Figure 1 cancers-14-02864-f001:**
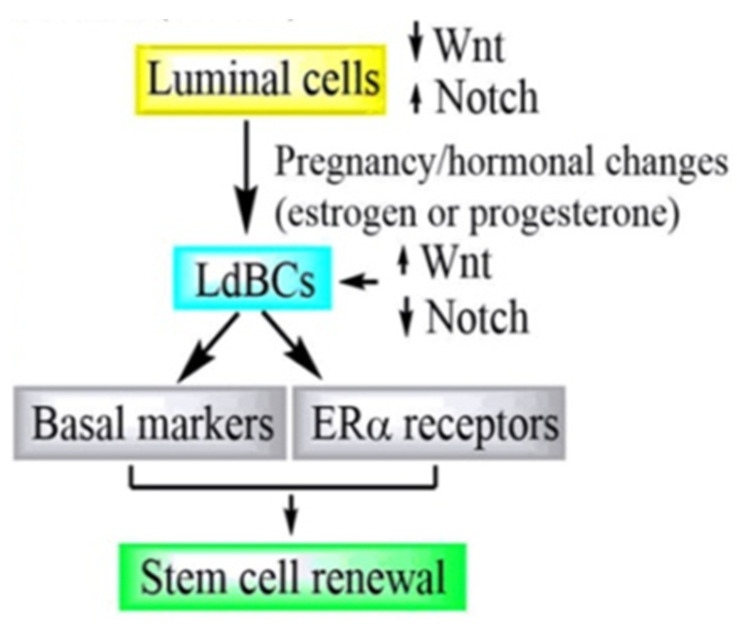
Sequential molecular events during self-renewal property acquisition in luminal A cells. LdBCs—Luminal-derived basal cells.

**Figure 2 cancers-14-02864-f002:**
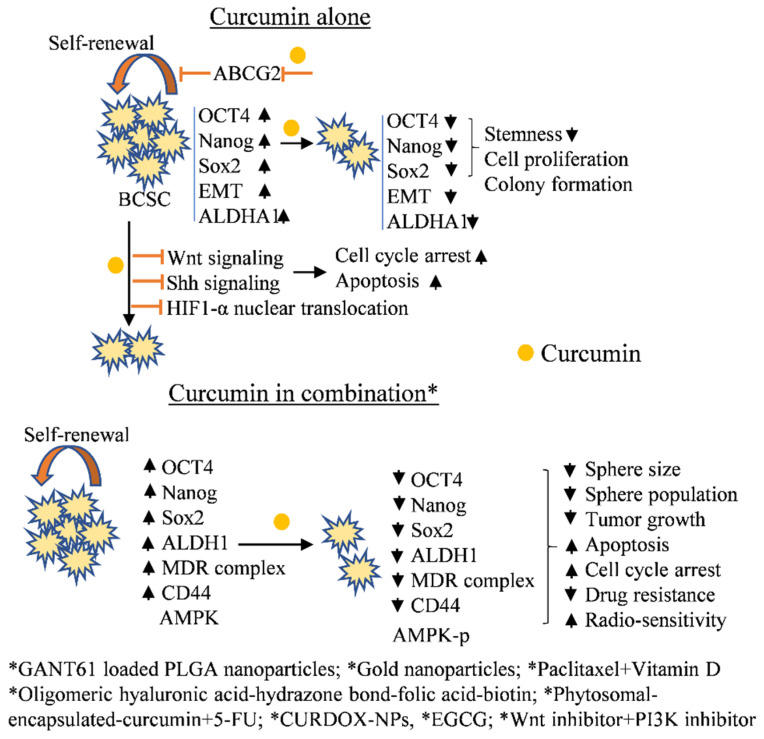
Effect of curcumin on luminal A-derived mammosphere. Yellow-colored circle represents curcumin. HIF-1α—Hypoxia inducing factor-1α, AMPK—AMP-activated protein kinase, EMT—Epithelial mesenchymal transition, Shh—Sonic hedgehog, ALDHA1—Aldehyde dehydrogenase A1, SRY (sex determining region Y)-box 2, and MDR—Multi-drug resistance.

**Figure 3 cancers-14-02864-f003:**
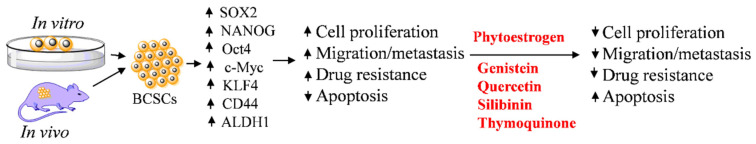
Effect of phytoestrogen(s) on the luminal A-derived mammosphere. ALDH1—Aldehyde dehydrogenase 1, Krüppel-like factor 4, and BCSCs—Breast cancer stem cells.

**Figure 4 cancers-14-02864-f004:**
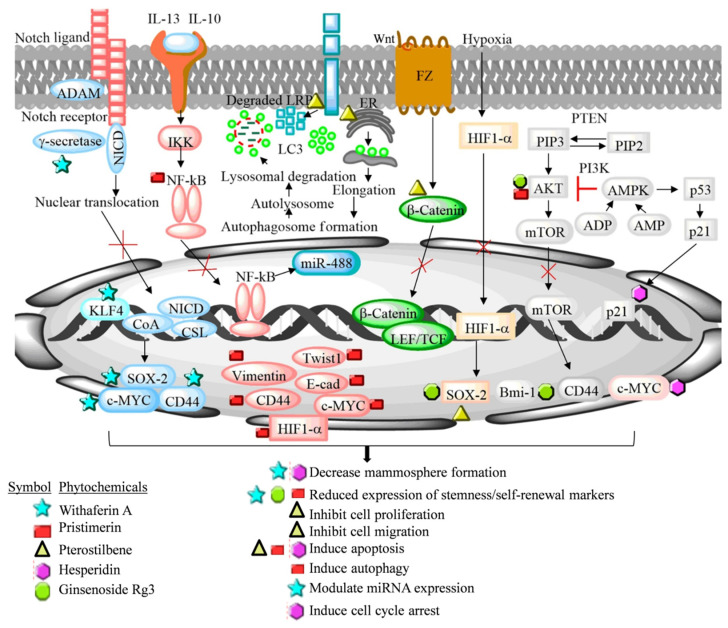
Luminal A-derived breast cancer stemness/self-renewal inhibiting potential of various dietary phytochemicals including *Withaferin A*, Ginsenoside Rg3 (GRg3), Pterostilbene, Hisperidin, and Pristimerin. The phytochemicals target Notch signaling, Wnt, hypoxia, NF-ҡB, and Akt pathways involved in the maintenance of stemness and self-renewal. Withaferin A reduces the mammosphere formation in luminal A breast cancer cells by modulating the expression of Sox2, c-MYC, and CD44 via targeting Notch signaling pathway. Pterostilbene inhibits NF-κB, vimentin, and Twist1, and elevated E-cadherin expression. Ginsenoside Rg3 inhibits the nuclear translocation of HIF-1α, Sox-2, and Bmi-1 marker expression via targeting hypoxia and PI3K/Akt signaling pathway. Pristimerin mediated modulation of autophagy and degradation of LRP6, a Wnt co-receptor of Wnt signaling pathway and the expression of stemness markers. Hesperidin arrest cell progression at G0/G1 phase by targeting cyclin D1. Hesperidin modulates the expression of p21 and p53 in luminal A-derived mammosphere. [KLF4—Kruppel-like factor 4, HIF-1α—Hypoxia inducing factor-1α, VEGFA—Vascular endothelial growth factor A, Hh—Hedgehog, PTCH1—Patched1, Smo—Smoothened, GLI1—Gli family zinc finger 1, Kinesin family member 7, FZ—Frizzled, TCF/LEF—T-Cell factor/Lymphoid enhance factor, IKB—Nuclear factor of kappa light polypeptide gene enhancer in B-cells inhibitor, mTOR—Mammalian target of rapamycin, PIP3—Phosphatidylinositol (3,4,5)-trisphosphate, PIP2—Phosphatidylinositol-4, 5-bisphosphate, PTEN—Phosphatase and tensin homolog, PI3K—Phosphoinositide 3-kinases, Akt—Protein kinase B, GREB—cAMP-response element binding protein, NICD—notch intracellular domain, HAT—Histone acetyltransferases; APH-1—anterior pharynx—defective 1, *PSEN*-Presenilin-1, NCSTN—Nicastrin, MDM2-*Mouse double minute* 2 homolog, Akt—Protein kinase B, SOX-2-(Sex determining region Y) box-2, Bmi-1—Polycomb protein complex, HIF-1α—Hypoxia inducing factor-1α, Wnt—Wingless/Integrated, LRP6—Low-density lipoprotein receptor-related protein 6, LC3-II—Light chain 3, NF-κB—Nuclear factor-kappa B, KLF4—Kruppel-like factor-4, AMPK—AMP-activated protein kinase, PI3K—Phosphoinositide-3-kinase, IL-6—Interleukin-6, mTOR—Mammalian target of rapamycin, P-gp—P-glycoprotein, YB-1—Y-binding protein 1, MDR—Multidrug resistance, AMP—Adenosine monophosphate, ADP—Adenosine diphosphate, PIP3—Phosphatidylinositol (3,4,5)-trisphosphate, PIP2—Phosphatidylinositol-4, 5-bisphosphate, PTEN—Phosphatase and tensin homolog, PI3K—Phosphoinositide 3-kinases, and Akt—Protein kinase B].

**Table 1 cancers-14-02864-t001:** List of dietary phytochemicals and associated secondary metabolite group targeting luminal A breast cancer cells.

Phytochemical Group	Phytochemicals	Reference
Isothiocynate	Benzyl Isothiocynate	[28]
Triterpene lactone, triterpenoids, monoterpene	Brusatol, Pristimerin, Thymoquinone	[29,30,31]
Phenolic, isoflavones, flavonoids, flavanone glycosid	Curcumin, Eugenol, Genistein, Pterostilbene, Quercetin, Silibinin, 6-Shogoal, Hesperidin, Quercetin-3-methyl ether	[30,32,33,34,35,36,37]
Anthraquinone	Emodin	[38]
Steroidal lactone, steroidal Saponin	Withaferin-A, Ginsenoside Rg3	[35,39]
Carbazole alkaloid	Mahanine	[40]

**Table 2 cancers-14-02864-t002:** Stemness/self-renewal signaling regulations by dietary phytochemicals in luminal A-derived BCaSCs.

Signaling Pathway	Markers for Validation	Regulatory Outcomes	Phytochemicals	Phytochemicals Effects	References
Akt	Sox-2, Bmi-1, HIF-1α	Mammosphere formation	Ginginoside Rg3	Decrease stemness/self-renewal	[25]
p53	p21, cyclin D1, p53	Mammosphere formation	Hesperidin	Reduce sphere formation, colony formation, migration, induce cell cycle arrest, apoptosis	[90]
Wnt	LRP6, p62 and LC3-II	Spheroid formation in BCa	Pristimerin	Inhibit self-renewal, induce apoptosis, autophagy	[91]
Hedgehog, Akt, β-catenin, Wnt, NF-κB	CD44 and c-Myc, β-catenin, HIF-1α, Twist1, Vimentin, E-cadherin, miR-448	Mammosphere formation	Pterostilbene	Reduce BCa stem cell generation, stemness related markers, metastasis, induce necrosis, sensitize chemotherapy	[92,93]
Notch4	uPAR, Bmi-1, KLF4, ALDH1	Stemness and self-renewal phenotype in BCaSCs	Withaferin A	Suppress stemness and self-renewal	[96]
Hypoxia, AMPK, STAT3, NF-κB	HIF-1*α,* HSP90, ARNT, Oct4, Nanog, Sox2, EMT, Bcl-2, ABCG2	Mammosphere formation, mainten cancer stem-like characteristics	Curcumin	Decrease cancer stemness/self-renewal markers expression, cell proliferation and colony formation, inhibit drug efflux transporters	[42,48,49,53,55]
Notch, Hedgehog, PI3K/Akt	DAXX, KLF4, IL-6	Expression self-renewal/stemness markers and side-population	Genistein	Inhibit mammosphere formation	[59,60,62,63,70]
PI3/AKT/mTOR	P-gp, YB-1	MDR in BCa cells, Cancer stem cell viability, mammosphere formation	Quercetin	Reduces BCaSCs’ proliferation, mammosphere generation, and colony formation	[74,79]
Metabolic pathway	Mitochondrial oxidative stress	Oxidative metabolism in BCaSCs	Silibinin	Reduces the sphere formation	[84]
	OCT-4, SOX-2, NANOG, ALDH1/2	Mammosphere formation	Thymoquinone	Self-renewal inhibition, mammosphere formation reduction	[87]

BCa—Breast cancer, BCaSCs—Breast cancer stem cells, Akt—Protein kinase B, SOX-2—(Sex determining region Y) box-2, Bmi-1—Polycomb protein complex, HIF-1α—Hypoxia inducing factor-1α, Wnt—Wingless/Integrated, LRP6—Low-density lipoprotein receptor-related protein 6, LC3-II—Light chain 3, NF-κB—Nuclear factor-kappa B, uPAR—Urokinase type plasminogen activator receptor, KLF4—Kruppel-like factor-4, ALDH1—Aldehyde dehydrogenase 1, AMPK-AMP—activated protein kinase, STAT3—Signal transducer and activator of transcription 3, HSP90—Heat shock protein 90, ARNT—Aryl hydrocarbon receptor nuclear transporter, Oct4—Octamer binding transcription factor, EMT—Epithelial mesenchymal transition, ABCG2—ATP-binding cassette super family G member 2, PI3K—Phosphoinositide-3-kinase, DAXX—Death-associated protein 6, IL-6—Interleukin-6, mTOR—Mammalian target of rapamycin, P-gp—P-glycoprotein, YB-1—Y-binding protein 1, and MDR—Multidrug resistance.

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
