# Peer review of "Targeting Breast Cancer-Derived Stem Cells by Dietary Phytochemicals: A Strategy for Cancer Prevention and Treatment"

_cancers, 2022, doi:10.3390/cancers14122864_

Round 1
Reviewer 1 Report
All suggested modifications have been made.
Author Response
We thank the reviewer for the positive comment as suggested modifications are made in the manuscript. There are no further comments.
Reviewer 2 Report
Addition of section 8 looks good to me. Few more concerns need to be addressed.
- Line 100, As “Positive response” only appeared once in the review, not necessary to provide abbreviation PR which will confuse with progesterone receptor (PR).
- Line 98-100. It’s unclear what does positive response (PR) > 20% refer to. No such info was found in your reference [11]. The reference states “….IHC-based lumina A tumors had better disease-free survival (DFS) if (progesterone receptor )PR was >20%.
- Luminal A refers to a subtype of breast cancer rather than a specific cell type(e.g. T cells, B cells). Using Lumina A cells-derived BCSC may be misleading and should be revised to Luminal A-derived Breast cancer stem cell like you did in line 104.
Author Response
Comment#1. Line 100, As “Positive response” only appeared once in the review, not necessary to provide abbreviation PR which will confuse with progesterone receptor (PR).
Response#1. The abbreviation is removed.
Concern#2. Line 98-100. It’s unclear what does positive response (PR) > 20% refer to. No such info was found in your reference [11]. The reference states “….IHC-based luminal A tumors had better disease-free survival (DFS) if (progesterone receptor PR was >20%.
Response#2. We appreciate the reviewer for careful review of the manuscript. The sentence has been modified.
Concern#3. Luminal A refers to a subtype of breast cancer rather than a specific cell type (e.g. T cells, B cells). Using Lumina A cells-derived BCSC may be misleading and should be revised to Luminal A-derived Breast cancer stem cell like you did in line 104.
Response#3. The point is well taken and the manuscript has been revised accordingly.